# Bilobalide Suppresses Adipogenesis in 3T3-L1 Adipocytes via the AMPK Signaling Pathway

**DOI:** 10.3390/molecules24193503

**Published:** 2019-09-27

**Authors:** Su Bu, Chun Ying Yuan, Quan Xue, Ying Chen, Fuliang Cao

**Affiliations:** 1College of Biology and the Environment, Nanjing Forestry University, Nanjing 210037, China; yuancy100294@outlook.com (C.Y.Y.); iyiudty@gmail.com (Q.X.); chynjfu@163.com (Y.C.); 2Co-innovation Center for Sustainable Forestry in Southern China, Nanjing Forestry University, Nanjing 210037, China

**Keywords:** bilobalide, 3T3-L1 cell, lipid metabolism, AMPK pathway

## Abstract

Bilobalide, the only sesquiterpene compound from *Ginkgo biloba* leaf, exhibits various beneficial pharmaceutical activities, such as antioxidant, anti-inflammation, and protective effects for the central nervous system. Several bioactive components extracted from *Ginkgo biloba* extract reportedly have the potential to attenuate lipid metabolism. However, the effect of bilobalide on lipid metabolism remains unclear. In this study, we used 3T3-L1 cells as the cell model to investigate the effect of bilobalide on adipogenesis. The results showed that bilobalide inhibited 3T3-L1 preadipocyte differentiation and intracellular lipid accumulation. Quantitative real-time PCR and western blotting results indicated that several specific adipogenic transcription factors and a few important adipogenesis-related genes were significantly down regulated on both mRNA and protein levels in bilobalide treatment groups. By contrast, the expression of some lipolytic genes, such as adipose triglyceride lipase, hormone-sensitive lipase (*HSL*), and carnitine palmitoyltransferase-1α, were all up-regulated by bilobalide treatment, and the phosphorylation of AMP-activated protein kinase (AMPK), acetyl-CoA carboxylase 1, and HSL were stimulated. Furthermore, bilobalide treatment partially restored AMPK activity following its blockade by compound C (dorsomorphin). These results suggested that bilobalide inhibited adipogenesis and promoted lipolysis in 3T3-L1 cells by activating the AMPK signaling pathway.

## 1. Introduction

Obesity is becoming a serious health problem in many countries and poses a major economic challenge worldwide. Obesity has transformed from a disease with high incidence in economically developed regions to a global chronic disease [1,2,3,4] that can lead to a number of serious complications, including type 2 diabetes (T2D), metabolic syndrome, some types of cancer, neurodegeneration, and cardiovascular diseases [5,6]. The proliferation of preadipocytes and lipid accumulation in adipocytes are the main causes of obesity [7].

Bioactive components from *Ginkgo biloba* have long been the focus of attention due to their high levels of pharmaceutical activities [8,9]. Bilobalide is a sesquiterpene compound (Figure 1) and the only one found in *Ginkgo biloba* [10]. Sesquiterpenes are one kind of important raw material used by the pharmaceutical industry and are ubiquitous in nature. Bilobalide is an intermediate metabolite of ginkgolides and is capable of inhibiting apoptosis, protecting injured cells, and protecting the brain from damage [11,12,13,14]. Ginkgolide A (GA) has been implicated in mitochondrial oxidative stress and inducing cellular lipoapoptosis [15], and also ginkgolide C reportedly exerts an inhibitory effect on lipid accumulation in liver hepatocellular carcinoma (HepG2) cells [16]. As the analogue of ginkgolides C and B, bilobalide might exhibit similar biological activity; therefore, in this study, we explored its effect on lipid metabolism.

Lipid storage in white adipose tissue (WAT) is controlled by the dynamic processes of lipogenesis and lipolysis, which involve many transcription factors, enzymes, and proteins. Among them, CCAAT enhancer binding protein alpha (C/EBPα), sterol regulatory element binding protein 1c (SREBP-1c), and peroxide proliferative activation receptor gamma (PPARγ) are key transcription factors responsible for end-phase differentiation and regulation of a large number of downstream genes involved in lipid metabolism [17,18]. AMP-activated protein kinase (AMPK) signaling is the key pathway of lipid metabolism. The β oxidation of fatty acids, lipid hydrolysis of triglycerides, and lipid production by adipocytes are regulated by the AMPK pathway [19]. Activation of this pathway requires AMPK phosphorylation, resulting in inhibition of lipid synthesis and upregulation of lipid hydrolysis and β oxidation of fatty acids [20]. Some proteins located downstream of the AMPK pathway, including acetyl coenzyme A carboxylase 1 (ACC1), sterol regulatory element binding protein 1c (SREBP-1c), glucose transporter 4 (GLUT-4), and carnitine palmitoyltransferase 1A (CPT1α), are all highly involved in lipid metabolism. ACC1 is a key rate-limiting enzyme in the first stage of fatty acid synthesis and promotes the synthesis of long-chain fatty acids, as well as the synthesis of triglycerides and phospholipids [21]. SREBP-1c increases intracellular cholesterol concentrations by enhancing the expression of cholesterol-binding receptors on the cell membrane surface [20]. GLUT-4, mainly found in fat and muscle tissue, regulates the uptake of glucose from extracellular cells after insulin stimulation [22]. CPT1α is located in the outer membrane of mitochondria and catalyzes the β oxidation of fatty acids to reduce the concentration of intracellular fatty acids [23]. Additionally, perilipin A located on the surface of lipid droplets in adipocytes prevents lipid hydrolysis of lipid droplets and protects them in adipocytes [24], whereas triglyceride lipase (ATGL) and hormone-sensitive lipase (HSL) are important to the process of fat hydrolysis and represent 95% of the activity of lipid-hydrolytic enzymes in adipose tissue [25]. All of these factors play key roles in lipid metabolism.

3T3-L1 preadipocytes are typical cells that can be specifically induced to differentiate into adipocytes and frequently used in lipid metabolism research. In this study, we explored the effects of bilobalide on adipogenesis and AMPK signaling in 3T3-L1 cells.

## 2. Results

### 2.1. Cell Cytotoxicity of Bilobalide on 3T3-L1 Cells

A thiazolyl blue tetrazolium bromide (MTT) assay was used to determine the cytotoxicity of bilobalide on both 3T3-L1 preadipocytes and differentiated adipocytes. Following treatment for 24 h, bilobalide at various concentrations did not show an inhibitory effect on 3T3-L1 preadipocyte proliferation but increased the cell viability slightly with statistical significance (Figure 2A), whereas bilobalide treatment exhibited significant cytotoxicity on mature adipocytes in a dose-dependent manner (Figure 2B). Compared with the control group, the cell viability of mature adipocytes significantly decreased to 67% following treatment with 200 μM bilobalide.

### 2.2. Bilobalide Inhibits Lipid Accumulation in 3T3-L1 cells

To evaluate the effect of bilobalide on adipogenesis, 3T3-L1 preadipocytes were treated with bilobalide at various concentrations during the process of adipo-differentiation for 5 days (day 4 to day 8). The intracellular lipid accumulation was evaluated by Oil Red O (ORO) staining. Microscopy showed that 25 μM and 100 μM bilobalide significantly decreased ORO staining in differentiated 3T3-L1 adipocytes (Figure 3A). We used isopropanol to extract the ORO dye and semi-quantitatively analyzed the absorbance at 510 nm, revealing that 25 μM and 100 μM bilobalide significantly inhibited the lipid accumulation by 25% and 39%, respectively, relative to the control group (*p* < 0.001) (Figure 3B).

### 2.3. Bilobalide Downregulates the Expression of Adipogenic Transcription Factors

Differentiation-associated lipid accumulation was induced by upregulated levels of several adipogenic transcription factors, including C/EBPα, PPARγ, and SREBP-1c. To evaluate the effect of bilobalide on their expression, 3T3-L1 preadipocytes were treated with 0 (control), 25, and 100 μM bilobalide during adipo-differentiation from day 4 to day 8. The results showed that mRNA levels of the three transcription factors were attenuated in a concentration-dependent manner. The 25 μM bilobalide decreased the mRNA levels of *PPARγ*, *C/EBPα*, and *SREBP-1c* by 30%, 18%, and 38%, respectively, compared to the control groups (*p* < 0.001); and the 100 μM bilobalide further decreased them by 57%, 38%, and 67%, respectively (Figure 4A, *p* < 0.001). In line with that, the protein expression of PPARγ, C/EBPα, and SREBP-1c went down steadily with the increasing dose of bilobalide. We observed a reduction of 80%, 25%, and 40%, respectively, in their protein levels upon the 100-μM bilobalide treatment, as compared with the control groups (Figure 4B).

### 2.4. The Effect of Bilobalide on the Expression of Adipogenic Genes in Adipocytes

We then evaluated whether bilobalide could attenuate the expression of several important adipogenesis-related genes, including fatty acid synthase (*FASN*), perilipin A, and *GLUT-4*. After the 25-μM bilobalide treatment, the mRNA levels of *FASN*, perilipin A, and *GLUT-4* decreased by 61%, 25%, and 83%, respectively, relative to controls (Figure 5A, *p* < 0.001). Although the 100-μM bilobalide treatment did not further decrease the transcription levels of those genes, their protein levels were continuously down-regulated with the increasing concentration of bilobalide (Figure 5A). The bilobalide treatment significantly reduced the protein expression levels of FASN and perilipin A by 22% (*p* < 0.01) and 51% (*p* < 0.001), respectively, at 25 μM, and by 37% (*p* < 0.05) and 91% (*p* < 0.001), respectively, at 100 μM. The 28% reduction in GLUT-4 protein by the 25-μM bilobalide was not statistically significant, while the 100-μM bilobalide gave rise to a 77% decrease with statistical significance (*p* < 0.001).

### 2.5. The Effect of Bilobalide on the Expression of Lipolytic Genes in 3T3-L1 Cells

We found that *ATGL* mRNA levels were up-regulated by 165% (*p* < 0.001) and 54% (*p* < 0.01) following the 25-μM and 100-μM bilobalide treatment, respectively, relative to controls, whereas *CPT1α* mRNA levels showed only a slight increase (~20%) at both concentrations (*p* < 0.05) (Figure 6A). Additionally, the protein levels of ATGL in the treatment groups were also increased by 47% (*p* < 0.01, 25 μM) and 57% (*p* < 0.001, 100 μM) as compared with that of the control groups, while the protein expression of CPT1α showed an upward trend with no statistical significance after the bilobalide treatment. Furthermore, the ratios of the phosphorylated HSL (the activated form) to the total HSL were raised by the bilobalide treatment and the 25-μM bilobalide treatment led to the optimal increase (Figure 6B, *p* < 0.001). These results indicated that bilobalide promoted the expression of genes involved in lipolysis.

### 2.6. Bilobalide Activates AMPK Signaling

Western Blot results indicated that the phosphorylation of AMPK and ACC1 was significantly enhanced by bilobalide at both 25 μM and 100 μM, with a 10-fold increase in the ratio of the phosphorylated ACC1 to the total ACC1 upon the 25-μM bilobalide treatment and a 158% increase in the ratio of the phosphorylated AMPK to the total AMPK upon the 100-μM bilobalide treatment (Figure 7A, *p* < 0.05). Additionally, enzyme-linked immunosorbent assay (ELISA) results showed similar relative AMPK activity at both bilobalide concentrations to that elicited by the treatment with 5-aminoimidazole-4-carboxamide ribonucleotide (AICAR; an activator of AMPK signaling). Furthermore, we confirmed that the treatment of compound C (an inhibitor of AMPK signaling) significantly attenuated the relative AMPK activity, whereas bilobalide at either concentration rescued the compound C-mediated inhibition (*p* < 0.001) (Figure 7B).

## 3. Discussion

Excessive eating and a sedentary lifestyle resulting in energy imbalance eventually leads to obesity, which is characterized by hypertrophy and hyperplasia of adipose tissue. Adipose tissue comprises adipocytes, preadipocytes, fibroblasts, vascular endothelial cells, and macrophages. Differentiation of preadipocytes into adipocytes (adipogenesis) is regulated by complex processes, which include the expression of adipogenic genes and activation of specific transcriptional factors and lipogenic enzymes. Because adipogenesis plays a key role in the development of obesity, it represents a target for obesity treatment, with current research focused on the discovery of safe and novel natural compounds capable of alleviating this process and to be used as alternative anti-obesity treatments [26,27,28].

*Ginkgo biloba* extract (GBE) reportedly exhibits extensive pharmaceutical activities [29]. Studies show that GBE can ameliorate lipid metabolism in high-fat diet-induced obese, diabetic, or non-alcoholic fatty liver disease animal models [30,31]. Additionally, several flavonoid components in GBE, including quercetin, kaempferol, and isorhamnetin, are widely found in fruits and vegetables and reportedly exhibit beneficial effects toward improving lipid profiles [32,33,34]. Ginkgolide A is non-toxic at high concentrations in hepatocytes, which inhibits lipid accumulation and displays hepatoprotective efficacy by inducing cellular lipoapoptosis and inhibiting cellular inflammation [15]. Ginkgolide C increases the expression of lipolytic enzymes, reduces adipogenic gene expression, and inhibits lipid accumulation in 3T3-L1 adipocytes [35].

Due to the structural similarity of bilobalide to ginkgolides, we hypothesized that bilobalide might also exert a regulatory effect on lipid metabolism. Here, we investigated the effect of bilobalide on lipid metabolism in 3T3-L1 cells. We found that bilobalide treatment reduced the accumulation of lipid droplets, inhibited the expression of adipogenic transcription factors and enzymes, and promoted AMPK phosphorylation to increase lipolysis in differentiated adipocytes.

AMPK is a metabolic energy sensor that plays a key role in regulating energy homeostasis [36,37]. Phosphorylation of AMPK stimulates substrate phosphorylation and inactivates the cytosolic isoform of ACC1 that provides malonyl-CoA the substrate necessary for fatty acid biosynthesis, thereby inhibiting fatty acid synthesis [38,39].

In the present study, we observed that bilobalide treatment promoted AMPK and ACC1 phosphorylation (Figure 7A). Our ELISA results verified that bilobalide promoted the relative AMPK activity to a degree similar to that elicited by 5-amino-4-imidazole carboxamide riboside (AICAR) (*p* < 0.05) (Figure 7B). Additionally, the AMPK phosphorylation up-regulated *ATGL* expression and phosphorylation of HSL (Figure 6). ATGL catalyzes the first step of triglycerol degradation to diacylglycerol and one molecule of free fatty acid [40], and phosphorylated HSL hydrolyzes diacylglycerol into monoacylglycerol and one molecule of fatty acid [41], with both of these activities enhancing hydrolysis of intracellular triglycerides [42]. Moreover, CPT1α activity is regulated by AMPK, which promotes *CPT1α* expression and accelerates the rate of fatty acid β oxidation in mitochondria [43]. Furthermore, SREBP-1c is a lipogenic transcription factor directly phosphorylated by AMPK [44] and is involved in inducing the expression of *ACC1* and *FASN* to stimulate fatty acid synthesis in cells [45]. Several studies demonstrated that phosphorylation of AMPK suppressed the ACC1 and FASN expression [46,47,48]. In the present study, we showed that bilobalide significantly suppressed the expression of adipogenic factors, such as *SREBP-1c*, *PPARγ*, *C/EBPα*, *FASN*, and perilipin A, during differentiation of 3T3-L1 cells. Although it is well established that AMPK activation will improve the translocation of GLUT4 to the cell membrane and improve the insulin sensitivity and whole body glucose homeostasis, a recent report indicated that mice overexpressing GLUT4 in adipocytes (AG4OX) have elevated adipose tissue lipogenesis and enhanced glucose tolerance despite being obese [49]. In accordance with our result of bilobalide treatment down-regulating GLUT4 expression and inhibiting adipogenesis, another study showed that pomegranate polyphenols and urolithin A down-regulated the gene expression of glucose and fatty acid metabolism, such as *GLUT4*, *FABP4*, and *PPARγ*, in 3T3-L1 like cells [50]. However, whether the down-regulation of GLUT4 in this study is AMPK-dependent or not requires further investigation.

Our MTT assay results indicated that bilobalide exhibited a dose-dependent toxicity against mature adipocytes but was non-toxic to preadipocytes. It is reasonable to speculate that bilobalide might trigger apoptosis in mature adipocytes, resulting in decreased adipose tissue. However, this hypothesis needs to be tested in an animal model.

In conclusion, our results demonstrated that bilobalide inhibited lipid accumulation in 3T3-L1 cells, reduced the expression of adipogenic genes, and enhanced lipolytic enzyme activity. Furthermore, our data suggested that AMPK signaling contributed to these effects (Figure 8). These findings represent the first evidence of bilobalide-mediated activation of AMPK signaling and its potential efficacy for the treatment of obesity. Studies are required to clarify the effect of bilobalide on fat accumulation in vivo.

## 4. Materials and Methods

### 4.1. Chemicals and Reagents

Dulbecco’s modified Eagle medium (DMEM), Dulbecco’s phosphate-buffered saline (DPBS), fetal bovine serum (FBS), and 0.25% trypsin were purchased from Gibco (Rockville, MD, USA). Penicillin-streptomycin solution was purchased from Hyclone (Provo, UT, USA). Insulin, dexamethasone (DEX), 3-isobutyl-1-methylxanthine (IBMX), and MTT were obtained from Sigma-Aldrich (St. Louis, MO, USA). 3T3-L1 preadipocytes were purchased from ATCC (Manassas, VA, USA). Bilobalide was obtained from the National Institutes for Food and Drug Control (Beijing, China). Primary antibodies specific for anti-pACC1 (S79), anti-ACC1, anti-FASN, anti-perilipin A, anti-ATGL, anti-C/EBPα, anti-PPARγ, anti-HSL, anti-pHSL (S563), anti-GLUT-4, anti-CPT-1α, anti-AMPK, and anti-pAMPK (T172) were acquired from Cell Signaling Technology (Danvers, MD, USA). Anti-SREBP-1c was acquired from Abcam (Cambridge, UK). Anti-β-Actin and goat anti-mouse and goat anti-rabbit IgG secondary antibodies were obtained from Boster Biological Technology (Pleasanton, CA, USA). The AMPK assay kit was purchased from Cell Signaling Technology.

### 4.2. Cell Culture

3T3-L1 preadipocytes were propagated in DMEM (low glucose) supplemented with 10% FBS and penicillin (100 U/mL), and streptomycin (100 μg/mL) and maintained in a humidified atmosphere of 5% CO_2_ at 37 °C. After reaching confluence, the medium was changed to DMEM (high glucose), and after 2 days (day 0), the medium was replaced with adipocyte-differentiation medium I (high-glucose DMEM containing 10% FBS, penicillin-streptomycin, 0.1 µM DEX, 0.5 mM 3-isobutyl-1-methylxanthine, and 10 μg/mL insulin). After 72 h (day 3), the medium was replaced with adipocyte-differentiation medium II (high-glucose DMEM containing 10% FBS, penicillin-streptomycin, and 10 μg/mL insulin). After 2 or 3 days of culture, the medium was replaced with DMEM containing 10% FBS and penicillin-streptomycin and cultured until the end of differentiation. Cells were treated with bilobalide at various concentrations from day 3 until the end of the differentiation.

### 4.3. MTT Assay

3T3-L1 cells were distributed in 96-well plates, and after reaching confluence or undergoing complete cell differentiation, the medium was replaced with 100 μL bilobalide at different concentrations and incubated for 24 h. Preadipocytes were treated with DMEM containing 10% FBS and corresponding concentrations of bilobalide, whereas adipocytes were treated with DMEM containing 1% FBS and corresponding concentrations of bilobalide. The culture medium was replaced with 50 μL of fresh culture medium containing 1 mg/mL MTT working solution, and after 4 h, formazan was dissolved in 100% dimethyl sulfoxide, and the absorbance was measured at 595 nm. Cell survivability (%) was expressed as the percentage of viable cells relative to the control.

### 4.4. ORO Staining

After differentiation, 3T3-L1 cells were washed twice with DPBS (pH 7.4), fixed with 10% formaldehyde for 1 h at room temperature, and stained with freshly prepared ORO working solution (0.3 mg/mL) at room temperature for 90 min, followed by two washes with distilled water. Images were obtained using a microscope (Leica DMIL LED; Leica, Frankfurt, Germany), and stained oil droplets were dissolved in isopropanol and quantified at 490 nm using a spectrophotometer (FilterMAX F5; Molecular Devices, USA).

### 4.5. Quantification of the Expression of Adipogenic Genes in 3T3-L1 Cells

Total RNA was extracted from 3T3-L1 cells using a MiniBEST universal RNA extraction kit (Takara Bio, Shiga, Japan). RNA concentrations were quantified using a NanoDrop ND-1000 Micro spectrophotometer (NanoDrop Technologies, Wilmington, DE, USA), and reverse transcription was performed using 1 μg of total RNA and PrimeScript RT master mix (Takara) according to the manufacturer’s instructions. qRT-PCR was performed using a SYBR Premix Ex Taq II kit (Takara), with the amplification reaction monitored on a qPCR thermal cycler (Step One Plus; Applied Biosystems, Foster City, CA, USA). Relative expression was determined using the 2*^−ΔΔCT^* method, with levels normalized against those of 18S rRNA or the housekeeping gene glyceraldehyde 3-phosphate dehydrogenase. Values are presented as fold changes compared to the control. The specific primers used in this study are listed in Table 1.

### 4.6. Western Blot

3T3-L1 adipocytes treated with the indicated concentrations of bilobalide during differentiation (days 3–9) were collected and lysed in radioimmunoprecipitation assay buffer (Biosharp, Anhui, China) containing 50 mM Tris (pH 7.4), 150 mM NaCl, 1% Triton X-100, 1% sodium deoxycholate, 0.1% SDS, a protease-inhibitor cocktail (Leupetin, PepstatinA, Aprotinin, E-64; Solarbio, Beijing, China), and a phosphatase-inhibitor cocktail (containing 25 μM (−)-p-bromotetramisole oxalate, 500 μM cantharidin, and 1 μM calyculin A; Beyotime, Shanghai, China). Total cellular proteins (30 μg) were separated by 10% SDS polyacrylamide gel electrophoresis (PAGE) or using 4% to 12% pre-cast SDS-PAGE gels (Bio-Rad, Hercules, CA, USA) and transferred onto nitrocellulose membranes (0.45 μm; Millipore, Bedford, MA, USA). The membranes were incubated overnight at 4 °C with primary antibodies, followed by three washes Tris-buffered saline containing 0.1% Tween-20 (Biosharp, HeFei, China), probing with horseradish peroxidase-conjugated secondary antibody, and development using enhanced chemiluminescence (ECL; Bio-Rad). The membranes were washed briefly with Tris-buffered saline (10 mM Tris-Cl and 150 mM NaCl; pH 7.5) supplemented with 0.05% Tween-20 (TBST), followed by blocking with TBST containing 5% (w/v) nonfat dried milk or 5% bovine serum albumin according to protocol associated with the respective primary antibody. Immunoreactive proteins were visualized using an ECL reagent (ECL Substrate; BioRad), and signals were quantified by densitometry with Image Lab software (BioRad).

### 4.7. AMPK Activity Assay

AMPK activity was determined using an AMPK kinase assay kit (Cell Signaling Technology) according to the manufacturer’s instructions. 3T3-L1 adipocytes were either incubated with 25 µM or 100 µM bilobalide or 40 µM AICAR for 24 h or pretreated with compound C (10 μM) for 1 h, followed by treatment with either bilobalide concentrations for 24 h. Cell lysates were extracted, as described previously, and transferred to the appropriate well containing a specific anti-AMPK primary antibody for overnight incubation at 4 °C. Lysates were then washed four times with wash buffer and incubated with secondary antibodies for 30 min at 37 °C, followed by the addition of 3,3′,5,5′-tetramethylbenzidine and incubation for 10 min at 37 °C, replacement with STOP solution, and determination of the absorbance at 450 nm within 30 min. Relative AMPK activity is presented as a percentage relative to the control.

### 4.8. Statistical Analysis

Values are expressed as the mean ± standard deviation. Significant differences between experiments performed at the same concentrations in treatment groups were determined using a Student’s *t* test (two-tailed).

## Figures and Tables

**Figure 1 molecules-24-03503-f001:**
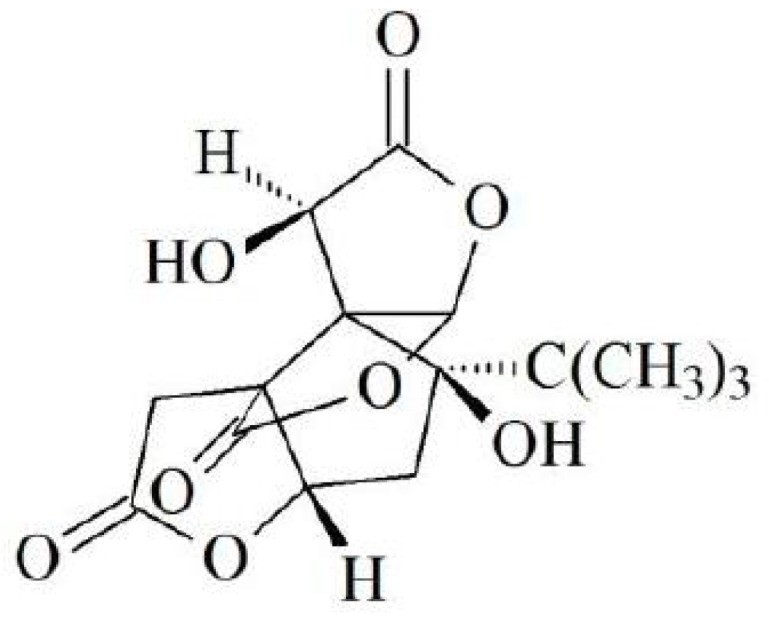
Chemical structure of Bilobalide.

**Figure 2 molecules-24-03503-f002:**
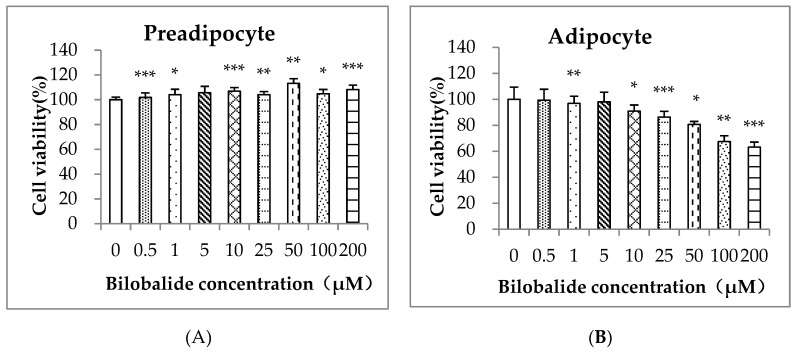
Cell viability of 3T3-L1 preadipocytes and adipocytes. (**A**) Preadipocytes and (**B**) adipocytes were treated with different concentrations (0–200 µM) of bilobalide for 24 h. Results are presented as the cell viability as compared with the compound-free group. Data are presented as the mean ± standard deviation from three independent experiments. * *p* < 0.05, ** *p* < 0.01, *** *p* < 0.001.

**Figure 3 molecules-24-03503-f003:**
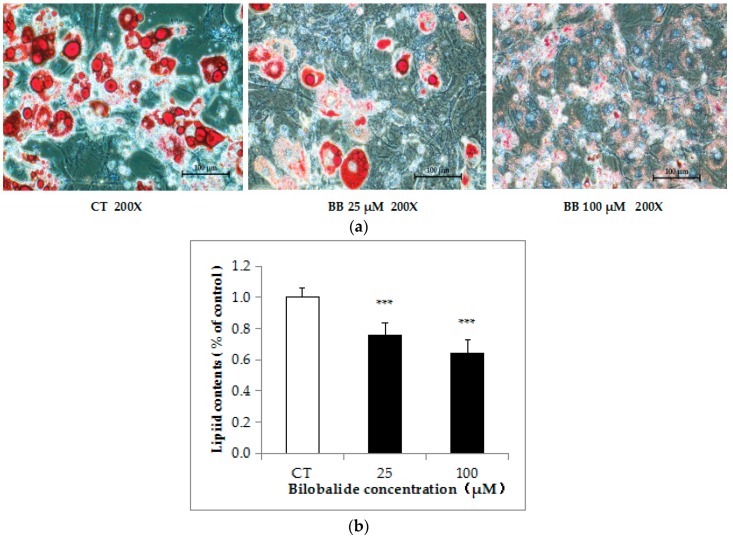
Effect of bilobalide on intracellular lipid accumulation during adipocyte differentiation. 3T3-L1 preadipocytes were cultured in adipo-differentiation media and treated with 0, 25, or 100 μM bilobalide and incubated for 5 days (day 4 to day 8) during differentiation, and staining was conducted using ORO solution. (**a**) Representative cell images were captured at 200 × magnification. (**b**) Intracellular lipid accumulation is expressed as a percentage of control values. Data are presented as the mean ± standard deviation ((*n* = 3). * *p* < 0.05, ** *p* < 0.01, *** *p* < 0.001.

**Figure 4 molecules-24-03503-f004:**
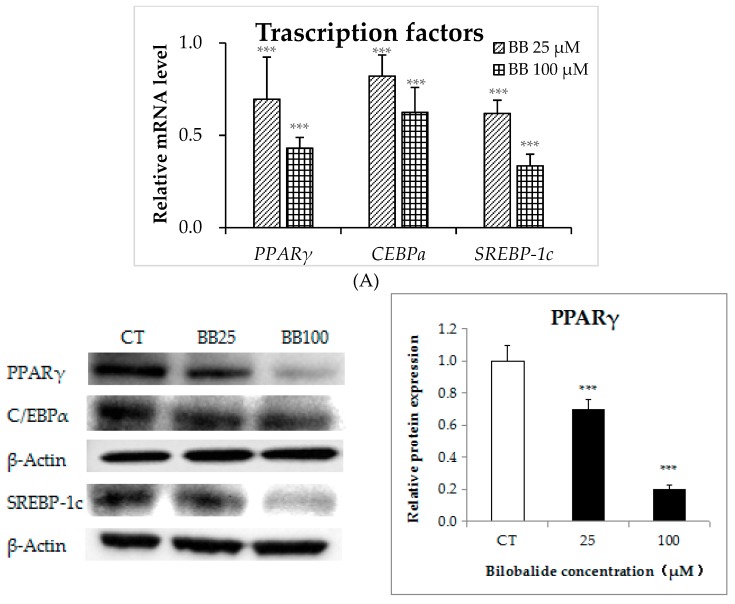
The effects of different bilobalide concentrations on adipogenic transcription factors. (**A**) The mRNA expression of *PPARγ*, *CEBP-α*, and *SREBP-1c*. (**B**) Western blot analysis of PPARγ, CEBP-α, and SREBP-1c protein levels. Histograms show quantification of protein levels according to band intensities. Data are presented as the mean ± standard deviation ((*n* = 3). * *p* < 0.05, ** *p* < 0.01, *** *p* < 0.001.

**Figure 5 molecules-24-03503-f005:**
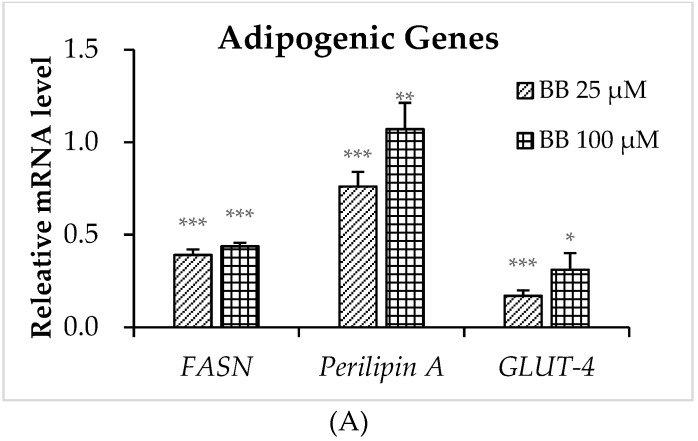
Effects of different bilobalide concentrations on the expression of adipogenic genes. 3T3-L1 cells were cultured 8 days after initiation of differentiation; cells were treated with 0, 25, or 100 µM bilobalide for 5 days during differentiation. (**A**) The relative mRNA expression of *FASN*, perilipin A, and *GLUT-4* was quantified by qRT-PCR. (**B**) Western blot was performed to determine protein levels of the same molecules. Histograms showed quantified levels of band intensities for each protein using Image Lab software. Data represent the mean ± standard deviation (n = 3). * *p* < 0.05, ** *p* < 0.01, *** *p* < 0.001.

**Figure 6 molecules-24-03503-f006:**
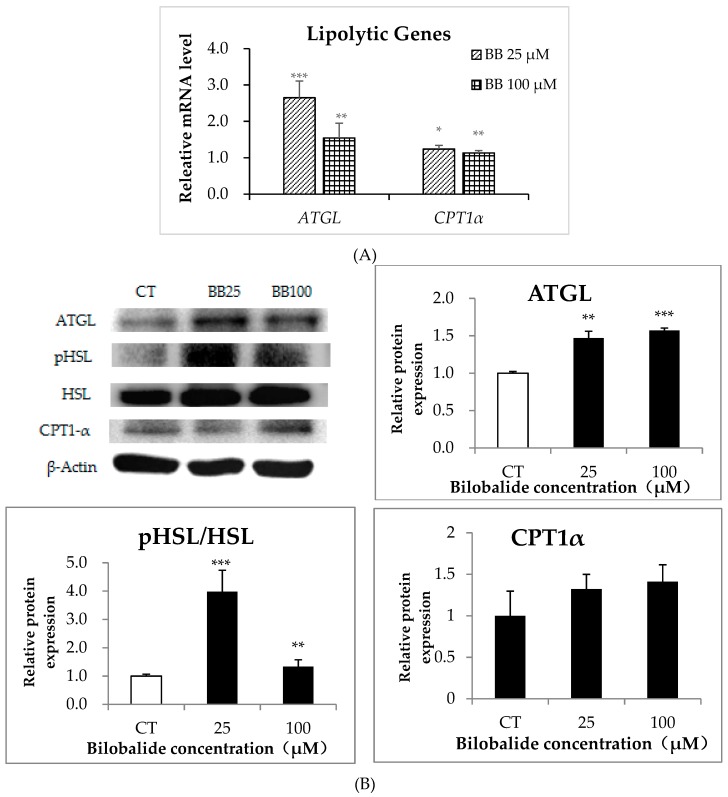
Effects of different bilobalide concentrations on the expression of lipolytic genes. 3T3-L1 cells were differentiated to adipocytes and treated with 0, 25, or 100 µM bilobalide for 24 h. (**A**) The relative mRNA expression of *ATGL*, *HSL*, and *CPT1**α* was quantified by qRT-PCR. (**B**) Western blot was performed to determine the protein levels of ATGL, HSL, phosphorylated HSL, and CPT1α. Protein bands were quantified by Image Lab software, and histograms represent quantified band intensities for each protein. Data represent the mean ± standard deviation (n = 3). * *p* < 0.05, ** *p* < 0.01, *** *p* < 0.001.

**Figure 7 molecules-24-03503-f007:**
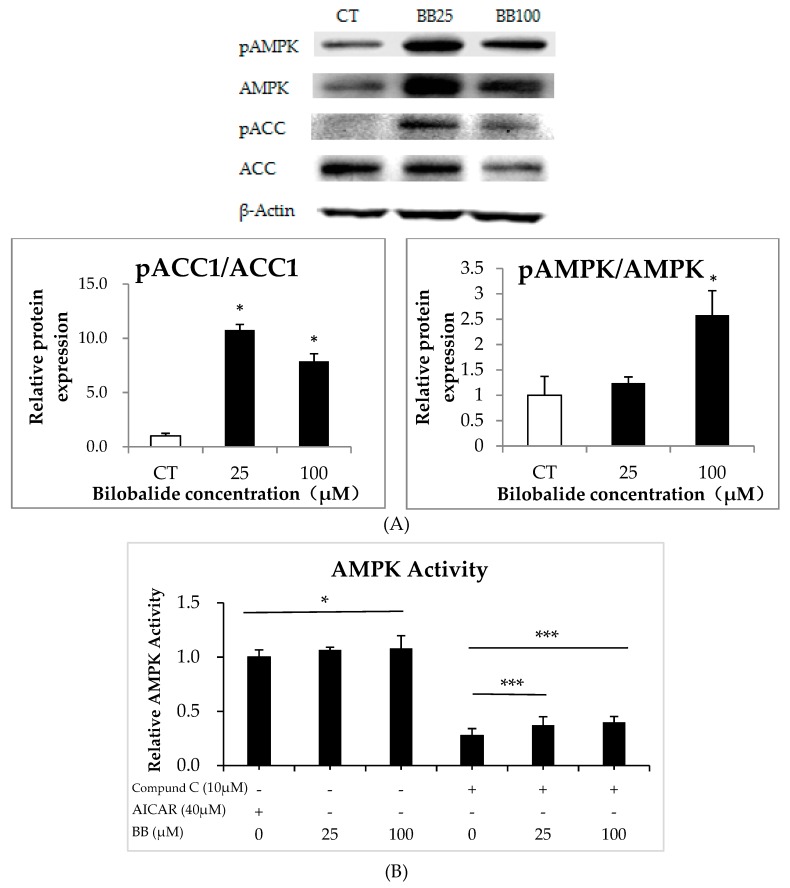
Bilobalide promotes AMPK phosphorylation and signaling. (**A**) Total protein was extracted from differentiated adipocytes, and changes in ACC1 and AMPK phosphorylation were determined. (**B**) Relative AMPK activity from ELISA assay following treatment with AICAR (40 μM), compound C (10 μM), and 25 μM and 100 μM bilobalide presented as a percentage relative to the control group. * *p* < 0.05, *** *p* < 0.001.

**Figure 8 molecules-24-03503-f008:**
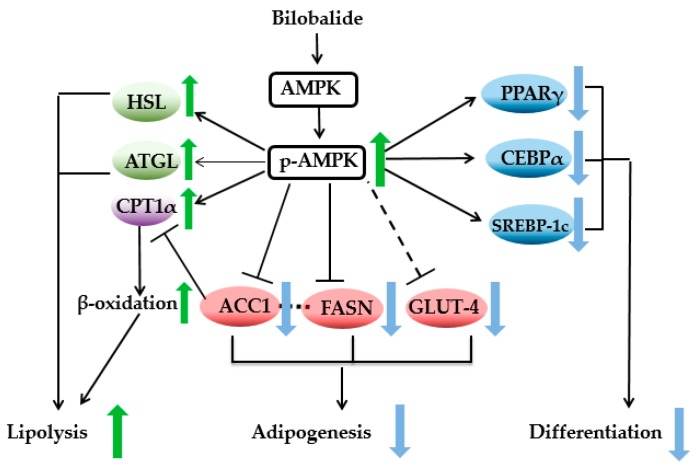
Schematic diagram showing the molecular mechanism of BB decreases lipid accumulation and inhibits obesity by down-regulating multiple transcription factors, inhibiting the synthesis of lipid accumulation proteins, and promoting the expression of lipid-degrading proteins in 3T3-L1 cells.

**Table 1 molecules-24-03503-t001:** Primers used for qRT-PCR.

Name	GenBank No.	Primer Sequence (5’– 3’)
*PPAR* *γ*	NM_001308354.1	F: AGGCCGAGAAGGAGAAGCTGTTG R: TGGCCACCTCTTTGCTCTGCTC
*C/EBPα*	NM_001308354.1	F: GCCCCCGTGAGAAAAATGAAG R: GAGGTGCGAAAAGCAAGGGA
*FASN*	NM_007988.3	F: ATTCGGTGTATCCTGCTGTC R: GCTTGTCCTGCTCTAACTGG
*SREBP-1c*	NM_001278601	F: TGGACTACTAGTGTTGGCCTGCTT R: ATCCAGGTCAGCTTGTTTGCGATG
*GLUT-4*	NM_009204	F: AGCCTCTGATCATCGCAGTG R: ACCGAGACCAACGTGAAGAC
*ATGL*	NM_001163689	F: GGTTAGAGTTGCTCAGCCGT R: ACATGAGGAGCGGATGTGTG
*CPT1α*	NM_013495	F: GGGCTTGGTAGTCAAAGGCT R: TGCCTGTGTCAGTATGCCTG
*Perilipin A*	NM_001113471.1	F: CACTCTCTGGCCATGTGGA R: AGAGGCTGCCAGGTTGTG
18S rRNA	NR_003278.3	F: ATGCGGCGGCGTTATTCC R: CTGTCAATCCTGTCCGTGTCC
*GAPDH*	XM_029476871.1	F: CAGGTTGTCTCCTGCGACTT R: TATGGGGGTCTGGGATGGAA

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
