# Peer review of "Bilobalide Suppresses Adipogenesis in 3T3-L1 Adipocytes via the AMPK Signaling Pathway"

_molecules, 2019, doi:10.3390/molecules24193503_

Round 1
Reviewer 1 Report
In this study, the authors found that bilobalide, an active component from Ginkgo biloba leaf, suppressed adipogenesis in 3T3-L1 preadipocytes via the AMPK signalling pathway. The results showed that bilobalide inhibited 3T3-L1 preadipocyte differentiation and intracellular lipid accumulation accompanied by down-regulation of adipogenic genes and proteins. However, the expression of some lipolytic genes such as ATGL, HSL, CPT-1αwere up-regulated with the treatment of bilobalide. It also showed that the effect of bilobalide on adipogenesis was associated with the activation of AMPK signalling. The reviewer has the following considerations.
3T3-L1 cell line is preadipocyte. The description of this cell line was not consistent in the text of the manuscript. In the paragraph “Cell cytotoxicity of bilobalide on 3T3-L1 cells” (results 2.1), the authors indicated that “Following treatment for 24 h, bilobalide at various concentrations showed no significant effect on 3T3-L1 preadipocyte viability”. However, in Figure 2(A), asterix indicating significance was shown. Some of the Western blot bands in Figures 4-7 were less clearly shown and the quantification of the density of the bands was lack of error bar in the bar chart. In Figure 4, the legends were not consistent with the presented panels. In Figure 8, according to the data presented, it seemed that the schematic diagram did not completely interpret the mechanisms of bilobalide on adipogenesis and lipolysis during adipogenic differentiation of 3T3-L1 cells. It will be informative to have the evidence that the phosphorylation of AMPK could be involved in the suppression of the expression of GLUT4, FASN and ACC1. In “Materials and Methods”, the information related with GLUT-4 and CPT-1α antibodies was lacking. It is suggested that multiple typos and grammar errors are corrected.Author Response
Please see the attachmen.

Reviewer 2 Report
Authors in this study demonstrated that bilobalide inhibited lipid accumulation in adipocytes, reduced the expression of adipogenic genes, and enhanced lipolytic enzyme activity controlled by AMPK signaling. These findings represent the first evidence of bilobalide-mediated activation of AMPK signaling and its potential efficacy for the treatment of obesity. This data could add up to our understanding of AMPK signaling in the context of obesity and could be of merit to treat its related diseases. I accept this study to be published in the esteemed journal
Minor comments:
In line 124-131, please provide the scale bar. Authors must provide statistical error bars for all western blot data. Figure 4B and 6A- please adjust the histograms properly.
Round 2
Reviewer 1 Report
The authors addressed the major considerations raised by the reviewer. Editing is suggested to make more accurate presentation of the results. And have consistent presentation of P value in the format e.g. P < 0.05 in whole manuscript.
